# Cryo-EM structure of the ATP-sensitive potassium channel illuminates mechanisms of assembly and gating

Gregory M Martin[1], Craig Yoshioka[2], Emily A Rex[1], Jonathan F Fay[1], Qing Xie[1], Matthew R Whorton[3], James Z Chen[1]*, Show-Ling Shyng[1]*

[1]Department of Biochemistry and Molecular Biology, Oregon Health and Science University, Portland, Oregon, United States; [2]Department of Biomedical Engineering, Oregon Health and Science University, Portland, Oregon, United States; [3]Vollum Institute, Oregon Health and Science University, Portland, Oregon, United States

**Abstract** $K_{ATP}$ channels are metabolic sensors that couple cell energetics to membrane excitability. In pancreatic $\beta$-cells, channels formed by SUR1 and Kir6.2 regulate insulin secretion and are the targets of antidiabetic sulfonylureas. Here, we used cryo-EM to elucidate structural basis of channel assembly and gating. The structure, determined in the presence of ATP and the sulfonylurea glibenclamide, at ~6 Å resolution reveals a closed Kir6.2 tetrameric core with four peripheral SUR1s each anchored to a Kir6.2 by its N-terminal transmembrane domain (TMD0). Intricate interactions between TMD0, the loop following TMD0, and Kir6.2 near the proposed $PIP_2$ binding site, and where ATP density is observed, suggest SUR1 may contribute to ATP and $PIP_2$ binding to enhance Kir6.2 sensitivity to both. The SUR1-ABC core is found in an unusual inward-facing conformation whereby the two nucleotide binding domains are misaligned along a two-fold symmetry axis, revealing a possible mechanism by which glibenclamide inhibits channel activity.

*For correspondence: chezh@ohsu.edu (JZC); shyngs@ohsu.edu (S-LS)

**Competing interests:** The authors declare that no competing interests exist.

## Introduction

Studies into the electric mechanisms of insulin release of the pancreatic $\beta$-cell in the early 1980s led to the discovery and identification of an ATP-sensitive potassium ($K_{ATP}$) channel as the key molecular link between glucose metabolism and insulin secretion (*Ashcroft and Rorsman, 1990*; *Cook and Bryan, 1998*). Subsequent cloning and characterization revealed the $\beta$-cell $K_{ATP}$ channel as a complex of two proteins: a potassium channel Kir6.2 of the inwardly rectifying $K^+$ channel family, and a sulfonylurea receptor SUR1, a member of the ATP binding cassette (ABC) transporter protein family (*Inagaki et al., 1995*).

Physiological activity of $K_{ATP}$ channels is determined primarily by the relative concentrations of ATP and ADP: ATP inhibits, whereas MgADP stimulates channel activity (*Nichols, 2006*). As $K_{ATP}$ channels set the $\beta$-cell membrane potential, this regulation by nucleotides endows them the ability to sense metabolic changes and translate those into changes in membrane excitability, which ultimately initiates or stops insulin secretion (*Ashcroft, 2005*). Another key player for $K_{ATP}$ function is membrane phosphatidylinositol-4, 5-bisphosphate ($PIP_2$); as in all other Kir family members, $PIP_2$ is required for channel opening and sets the intrinsic open probability ($P_o$) of the channel (*Hibino et al., 2010*; *Nichols, 2006*). Mutations disrupting channel assembly or the above-gating properties result in insulin secretion disorders, with loss- or gain-of-function mutations causing congenital hyperinsulinism (HI) or permanent neonatal diabetes mellitus (PNDM), respectively (*Ashcroft, 2005*). Importantly, $K_{ATP}$ channels are the targets of sulfonylureas, one of the most commonly

**eLife digest** The hormone insulin reduces blood sugar levels by encouraging fat, muscle and other body cells to take up sugar. When blood sugar levels rise following a meal, cells within the pancreas known as beta cells should release insulin. In people with diabetes, the beta cells fail to release insulin, meaning that the high blood sugar levels are not corrected.

When blood sugar levels are high, beta cells generate more energy in the form of ATP molecules. The increased level of ATP causes channels called ATP-sensitive potassium ($K_{ATP}$) channels in the membrane of the cell to close. This triggers a cascade of events that leads to the release of insulin.

Some treatments for diabetes alter how the $K_{ATP}$ channels work. For example, a widely prescribed medication called glibenclamide (also known as glyburide in the United States) stimulates the release of insulin by preventing the flow of potassium through $K_{ATP}$ channels. It remains unknown exactly how ATP and glibenclamide interact with the channel's molecular structure to stop the flow of potassium ions.

$K_{ATP}$ channels are made up of two proteins called SUR1 and Kir6.2. To investigate the structure of the $K_{ATP}$ channel, Martin et al. purified channels made of the hamster form of the SUR1 protein and the mouse form of Kir6.2, which each closely resemble their human counterparts. The channels were purified in the presence of ATP and glibenclamide and were then rapidly frozen to preserve their structure, which allowed them to be visualized individually using electron microscopy. By analyzing the images taken from many channels, Martin et al. constructed a highly detailed, three-dimensional map of the $K_{ATP}$ channel. The structure revealed by this map shows how SUR1 and Kir6.2 work together and provides insight into how ATP and glibenclamide interact with the channel to block the flow of potassium, and hence stimulate the release of insulin.

An important next step will be to improve the structure to more clearly identify where ATP and glibenclamide bind to the $K_{ATP}$ channel. It will also be important to study the structures of channels that are bound to other regulatory molecules. This will help researchers to fully understand how $K_{ATP}$ channels located throughout the body operate under healthy and diseased conditions. This knowledge will aid in the design of more effective drugs to treat several devastating diseases caused by defective $K_{ATP}$ channels.

prescribed treatments for type 2 diabetes, which stimulate insulin secretion by inhibiting channel activity (*Gribble and Reimann, 2003*). In particular, glibenclamide (GBC) binds the channel with nanomolar affinity and was instrumental for the purification and cloning of SUR1 (*Aguilar-Bryan et al., 1995*).

A member of the Kir channel family, Kir6.2 consists of two transmembrane helices and N- and C-terminal cytoplasmic domains (*Hibino et al., 2010*). By comparison, SUR1, a member of the ABC transporter family, is much larger in size. In addition to a characteristic ABC core structure comprising two transmembrane domains (TMD1 and 2) and two cytoplasmic nucleotide binding domains (NBD1 and 2), it has an N-terminal extension that contains a transmembrane domain (TMD0) followed by a long, cytoplasmic loop 'L0' which connects to the ABC core (*Aguilar-Bryan et al., 1995*; *Tusnády et al., 2006*). Kir6.2 and SUR1 are uniquely dependent on each other for expression and function (*Inagaki et al., 1995*). Interestingly, unlike most ABC transporters such as the cystic fibrosis transmembrane conductance regulator (CFTR) and the multidrug-resistant protein P-glycoprotein, SUR1 itself has no known ion channel or transporter activity; instead, its function is to regulate Kir6.2 channels (*Aguilar-Bryan et al., 1995*; *Inagaki et al., 1995*; *Wilkens, 2015*). A central question is how the two proteins assemble and function as a complex to sense metabolic signals.

Biochemical and biophysical studies have indicated that the $K_{ATP}$ channel is an octamer of four Kir6.2 and four SUR1 subunits. ATP and $PIP_2$ bind Kir6.2 directly to close or open the channel, respectively (*Baukrowitz et al., 1998*; *Shyng and Nichols, 1998*; *Tanabe et al., 1999*; *Tucker et al., 1997*). Although Kir6.2 alone can be gated by ATP and $PIP_2$, its sensitivities to both ATP and $PIP_2$ are increased by SUR1 by ~10-fold (*Baukrowitz et al., 1998*; *Enkvetchakul et al., 2000*; *Shyng and Nichols, 1998*; *Tucker et al., 1997*). How SUR1 sensitizes Kir6.2 to ATP inhibition and $PIP_2$ stimulation remains unclear. In contrast to ATP inhibition of the channel, which does not

depend on $Mg^{2+}$ and ATP hydrolysis, nucleotide stimulation of the channel is conferred by SUR1 and requires $Mg^{2+}$ (*Ashcroft and Gribble, 1998*; *Gribble et al., 1997*, *1998*; *Nichols, 2006*). Evidence suggests that MgATP and MgADP interact with the nucleotide binding domains (NBDs) of SUR1 and either through MgATP hydrolysis or through direct MgADP binding at NBD2, promote NBDs dimerization and channel opening (*de Wet et al., 2012*; *Nichols, 2006*; *Zingman et al., 2007*). Moreover, GBC has been proposed to inhibit $K_{ATP}$ channels by preventing Mg-nucleotide stimulation (*de Wet and Proks, 2015*), and may do so by stabilizing the ABC core of SUR1 in an inward-facing conformation (*Ortiz et al., 2012*) but direct evidence is lacking.

In order to understand how the channel functions as a complex to respond to physiological and pharmacological molecules and mechanisms by which channel mutations cause disease, detailed structural information is crucial. Here, we used cryo-EM to elucidate the structural basis of $K_{ATP}$ channel assembly and gating.

## Results

### Structure determination

To obtain sufficient quantity of purified channel complexes, we used rat insulinoma INS-1 cells, which naturally express $K_{ATP}$ channels, for overexpression. Cells were transduced with recombinant adenoviruses encoding genes for a FLAG-tagged hamster SUR1 and a rat Kir6.2 (*Pratt et al., 2009*), which are 95% and 96% identical to the human sequences, respectively. These heterologously expressed channels have gating properties indistinguishable from endogenous $K_{ATP}$ channels (*Pratt et al., 2009*). Channel integrity was found to be best preserved when membranes were solubilized in digitonin and channels purified in the presence of 1 μM glibenclamide (GBC) and 1 mM ATP (see Materials and methods) (*Figure 1*), which was the condition used for cryo-EM structure determination.

Single-particle analysis using RELION identified two three-dimensional (3D) classes of particles with distinct conformations in the cytoplasmic domain of Kir6.2 (see Discussion below). The dominant class (~60%) produced a reconstruction which has an overall resolution of 6.7 Å (FSC = 0.143) with C4 symmetry imposed (*Figure 1—figure supplements 1* and *2*; *Table 1*). With masking the FSC measurement at 0.143 reached 5.8 Å and the Kir6.2 core 5.1 Å. The other class yielded a reconstruction with an overall unmasked resolution ~7.6 Å, and masked whole channel and Kir6.2 core ~7.2 Å and 6.9 Å, respectively. The higher resolution map was used for model building and structural analysis. All transmembrane (TM) helices were clearly resolved in the density map (76 total; 17 from each SUR1, 2 from each Kir6.2; *Figure 2*) and contained significant side-chain density which allowed for registration of the models.

Kir6.2 is a member of the highly conserved Kir channel family in which several structures have been solved (*Hibino et al., 2010*). By contrast, SUR1 is one of the few ABC transporter proteins which have an N-terminal extension consisting of a transmembrane domain termed TMD0 followed by a long intracellular loop (the third intracellular loop, ICL3) termed L0, in addition to an ABC core structure comprising two transmembrane domains (TMD1 and 2) and two nucleotide binding domains (NBD1 and 2) (*Tusnády et al., 2006*). The Kir6.2 and SUR1 ABC core domain models were built initially from homologous Kir and ABC transporter structures (sequence and model comparisons with templates shown in *Figure 2—figure supplements 1–4*) and then refined to fit the density. Because there is no known structural template for the TMD0-L0 of SUR1, this region was modeled de novo.

### Overall architecture of the $K_{ATP}$ channel

The structure shows that the $K_{ATP}$ channel is an octamer built around a Kir6.2 tetramer with each subunit complexed to one SUR1 (*Figure 2*). The complex is ~200 Å in width in the longest dimension and ~125 Å in height, and is shaped like a propeller with the Kir6.2 pore and TMD0 forming a compact central core and the SUR1-ABC core structure forming the blades.

A long-standing question has been where TMD0 and L0 are in relation to Kir6.2 and the ABC core structure, as this region has been shown to be crucial for channel assembly and gating (*Babenko and Bryan, 2003*; *Chan et al., 2003*; *Schwappach et al., 2000*). An earlier model hypothesized TMD0 to be sandwiched between Kir6.2 and the TMDs of the ABC core (*Bryan et al., 2004*), but a later cryo-negative stain single-particle EM study of a channel formed by a SUR1-Kir6.2 fusion

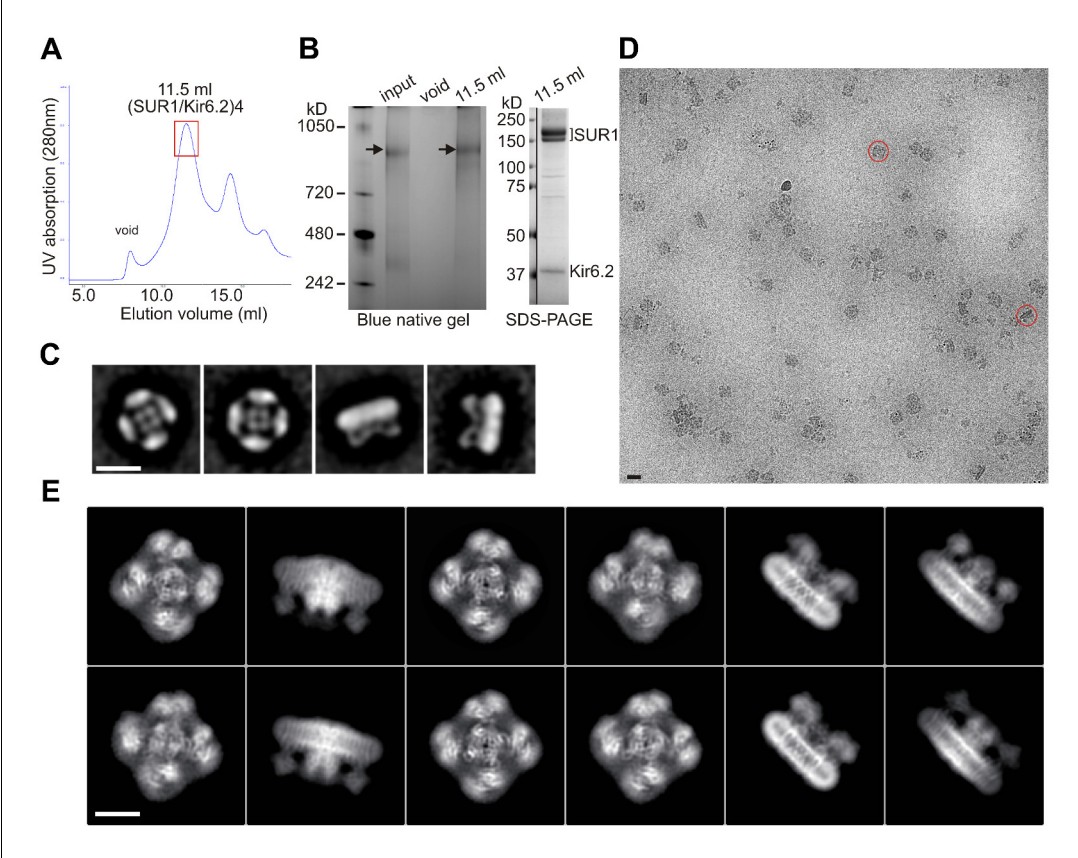

**Figure 1.** Purification and single-particle EM imaging of the SUR1/Kir6.2 $K_{ATP}$ channel. (A) Size exclusion chromatography (SEC) profile of affinity purified $K_{ATP}$ channels on a Suprose 6 column showing peak elution at ~11.5 ml (the red rectangle). (B) *Left*: Blue native gel showing the size of the purified complex at ~1 mDa (arrow) corresponding to four SUR1 and four Kir6.2. Input: samples eluted from anti-FLAG M2 agarose beads; void: sample from the SEC void fraction; 11.5 ml: sample from the SEC 11.5 ml elution fraction. *Right*: SDS-PAGE of the 11.5 ml fraction showing SUR1 (lower band: core-glycosylated; upper band: complex-glycosylated) and Kir6.2 as the main proteins. A vertical line separates MW markers from the sample lane in the same gel. (C) Negative-stain two-dimensional class averages showing topdown views (1, 2) and side views (3, 4) of the channel complex. (D) A representative cryoEM micrograph of $K_{ATP}$ channel particles imaged on an UltrAufoil grid. (E) Representative two-dimensional class averages of $K_{ATP}$ channels.

The following figure supplements are available for figure 1:

**Figure supplement 1.** Cryo-EM data processing flowchart.
**Figure supplement 2.** Cryo-EM density map analysis.

protein placed TMD0 next to Kir6.2 in between two adjacent SUR1-ABC core domains (*Mikhailov et al., 2005*). In our structure, TMD0-L0 sits in between the SUR1 and Kir6.2 subunits and is the primary point of contact between the SUR1-ABC core and Kir6.2 (*Figure 2*).

## The Kir6.2 tetramer is in a closed conformation

The Kir6.2 tetramer is the best resolved region in the complex (*Figure 3A*). Side-chain density of many residues, in particular those in the two TM helices are visible (*Figure 3B*). With knowledge of existing Kir channel structures, this allowed for confident model building (see Materials and methods; sequence comparison with the template is shown in *Figure 2—figure supplement 1*).

A vertical slice through the middle of the channel highlights the $K^+$ conduction pathway (*Figure 3C*). The three constriction points correspond to the selectivity filter, inner helix gate, and G-loop gate in other known Kir structures (*Hansen et al., 2011*; *Whorton and MacKinnon, 2011*).

**Table 1.** Statistics of cryo-EM data collection, 3D reconstruction and model building.

| Data collection/processing | |
|---|---|
| Microscope | Krios |
| Voltage (kV) | 300 |
| Camera | Gatan K2 |
| Camera mode | Counting |
| Defocus range (μm) | 1.2 ~ 3.5 |
| Exposure time (s) | 20 |
| Dose rate (e⁻/pixel/s) | 6 |
| Magnified pixel size (Å) | 1.72 |
| Total dose (e-/Å²) | 40 |
| **Reconstruction** | |
| Software | RELION |
| Symmetry | C4 |
| Particles refined | 27371 |
| Resolution (unmasked, Å) | 6.7 |
| Resolution (masked, Å) | 5.8 |
| Resoultion (Kir6.2 masked, Å) | 5.1 |
| Map sharpening B-factor (Å²) | −250 |
| **Model Statistics** | |
| Map CC | 0.95 (masked) |
| Resolution (FSC = 0.5, Å) | 5 Å (via phenix model-map FSC) |
| MolProbity score | 2.26 |
| Cβ deviations | 0 |
| **Ramachandran** | |
| Outliers | 0.12% |
| Allowed | 4.68% |
| Favored | 95.20% |
| **RMS deviations** | |
| Bond length | 0.005 |
| Bond angles | 1.262 |

In Kir6.2, the inner helix gate is formed by F168 in M2 just below the central cavity. In our model, there is only ~6 Å between opposing atoms of the gate (~3 Å when considering the van der Waals radii), which is too narrow to allow passage of a ~8 Å diameter hydrated $K^+$ ion (*Figure 3D*). The G-loop gate formed at the apex of the cytoplasmic domains is shown in *Figure 3E*. A comparison of closed (Kir3.2 apostate) and open (Kir3.2-R201A + $PIP_2$) G-loop structures in relation to Kir6.2 suggests that this gate is also closed (*Figure 3E*). Together, these observations indicate a closed channel structure, which is expected since the sample contained saturating concentrations of inhibitory ATP and GBC.

Interestingly, 3D classification identified two classes with distinct conformations in the cytoplasmic domain (CTD) of Kir6.2. The two classes differ by a rigid-body rotation of the CTD of ~14° (*Figure 1—figure supplement 2F*). A similar rotation has been observed in multiple Kir channel members and has been associated with channel gating (*Clarke et al., 2010*; *Whorton and MacKinnon, 2013*). However, the TMD and gates as well as the density corresponding to bound ATP (see below) in both classes are largely unaffected, suggesting rotational freedom for the CTD in the

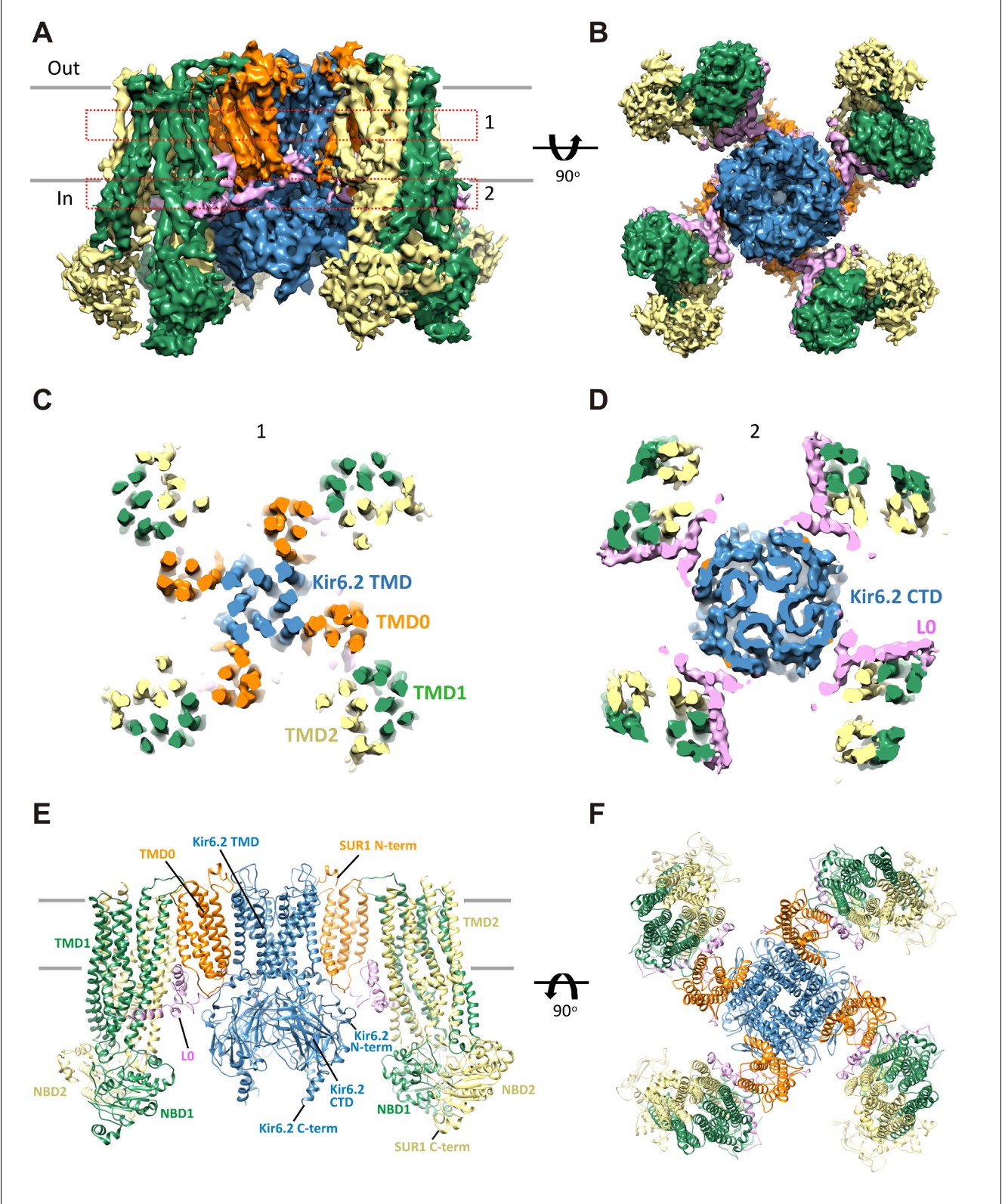

**Figure 2.** Three-dimensional reconstruction of the K$_{ATP}$ channel. (**A**) Cryo-EM density map of the K$_{ATP}$ channel complex at an overall resolution of 5.8 Å, viewed from the side. The four Kir6.2 subunits in the center are colored blue, SUR1 is in orange (TMD0), lavender (L0), green (TMD1/NBD1), and yellow (TMD2/NBD2). Gray bars indicate approximate positions of the lipid bilayer. (**B**) View of the complex from the cytoplasmic side. (**C** and **D**) Cross-

*Figure 2 continued on next page*

*Figure 2 continued*

sections of the density map. The planes where the sections 1 and 2 are made are shown in (A). (E) Model of SUR1 and Kir6.2 constructed from the EM density map viewed from the side. A Kir6.2 tetramer and only two SUR1 subunits are shown for clarity. (F) The model viewed from the extracellular side.

The following figure supplements are available for figure 2:

**Figure supplement 1.** Sequence and structure comparison between Kir6.2 and Kir3.2.

**Figure supplement 2.** Sequence alignment and structure comparison between SUR1 TMD1 and a bacterial peptidase-containing ABC transporter PCAT-1 (PDB ID: 4RY2).

**Figure supplement 3.** Sequence alignment and structure comparison between SUR1 NBD1 and the mouse P-glycoprotein NBD1 (PDB ID: 4MLM).

**Figure supplement 4.** Sequence alignment and structure comparison between SUR1 TMD2-NBD2 and a bacterial ABC exporter TM287/288 (PDB ID: 4Q4H).

closed state. Whether this rotation represents a conformational transition that occurs during gating needs further investigation.

## Identification of the ATP binding pocket

A hallmark of the $K_{ATP}$ channel is its inhibition by intracellular ATP. Mutagenesis and biochemical studies suggest that ATP binds directly to Kir6.2 (*Tanabe et al., 1999*; *Tucker et al., 1997*), and that residues in both N- and C-terminal domains are involved (*Antcliff et al., 2005*; *Nichols, 2006*). However, while Kir6.2 is sensitive to ATP in the absence of SUR1 ($IC_{50}$ ~100 μM), SUR1 increases this sensitivity by ~10-fold ($IC_{50}$ ~10 μM) (*Tucker et al., 1997*). Where ATP binds and how SUR1 enhances the sensitivity to ATP inhibition remain key questions.

Since our preparation contained 1 mM ATP, we reasoned that ATP is likely bound to the channel. Indeed, we observed a prominent bulge in the EM density that is too large to be accounted for by the main chain and the surrounding side-chains. The density is about the size of an ATP molecule and is immediately adjacent to K185, a residue that has been implicated in ATP binding (*John et al., 2003*; *Tanabe et al., 1999*; *Tucker et al., 1997*). Extensive mutagenesis of the K185 residue assessing the effects of various amino acid substitutions on channel sensitivity to inhibition by ATP, ADP, and AMP has provided strong evidence that this residue is important for binding to the β-phosphate of ATP (*Jöns et al., 2006*). We used this information to guide the initial docking of ATP into the density and then refined with the surrounding protein in RSRef (*Chapman et al., 2013*).

An overview of the ATP binding site from the side (*Figure 4A*), and from the top (*Figure 4B*), with ATP colored in red, illustrates that the pocket is at the interface of adjacent Kir6.2 N and C domains. A close-up view (*Figure 4C*) shows that the docked ATP is surrounded by residues I182, L205, Y330, F333, and G334 from the same subunit, and R50 from the adjacent subunit. The adenine ring is pointing toward the N-terminus of subunit A, and could be supported by I182, L205, Y330, and F333 of subunit B. R50 in subunit A is in a position that would allow it to interact with the γ-phosphate but may also interact with the adenine ring, which would explain mutagenesis data indicating that the interaction of R50 and ATP is not entirely electrostatic (*John et al., 2003*). K185 is only ~3 Å from the β phosphate, while the α-phosphate is close to the main-chain nitrogen of G334 (*Figure 4C,D*). Importantly, most residues surrounding the ATP density have been mutated and shown to affect ATP sensitivity (*Antcliff et al., 2005*), providing direct validation of our structure.

In our structure, we see that the density corresponding to ATP is located on the periphery of the Kir6.2 cytoplasmic domain, and traversed by the N-terminal segment of L0 of SUR1 immediately following TMD0 (*Figure 4B*), with the Cα of K205 coming within only ~10 Å of the site (*Figure 4D*). Interestingly, we have previously shown that mutation of K205 of L0 to alanine or glutamate reduce ATP sensitivity by ~10-fold (*Pratt et al., 2012*). While there is no density in the map to allow placement of the K205 side chain, its Cα position lies directly over the site and is poised to make electrostatic contribution to ATP binding. This finding offers a mechanism by which SUR1 could enhance the ATP-sensitivity of the Kir6.2 channel.

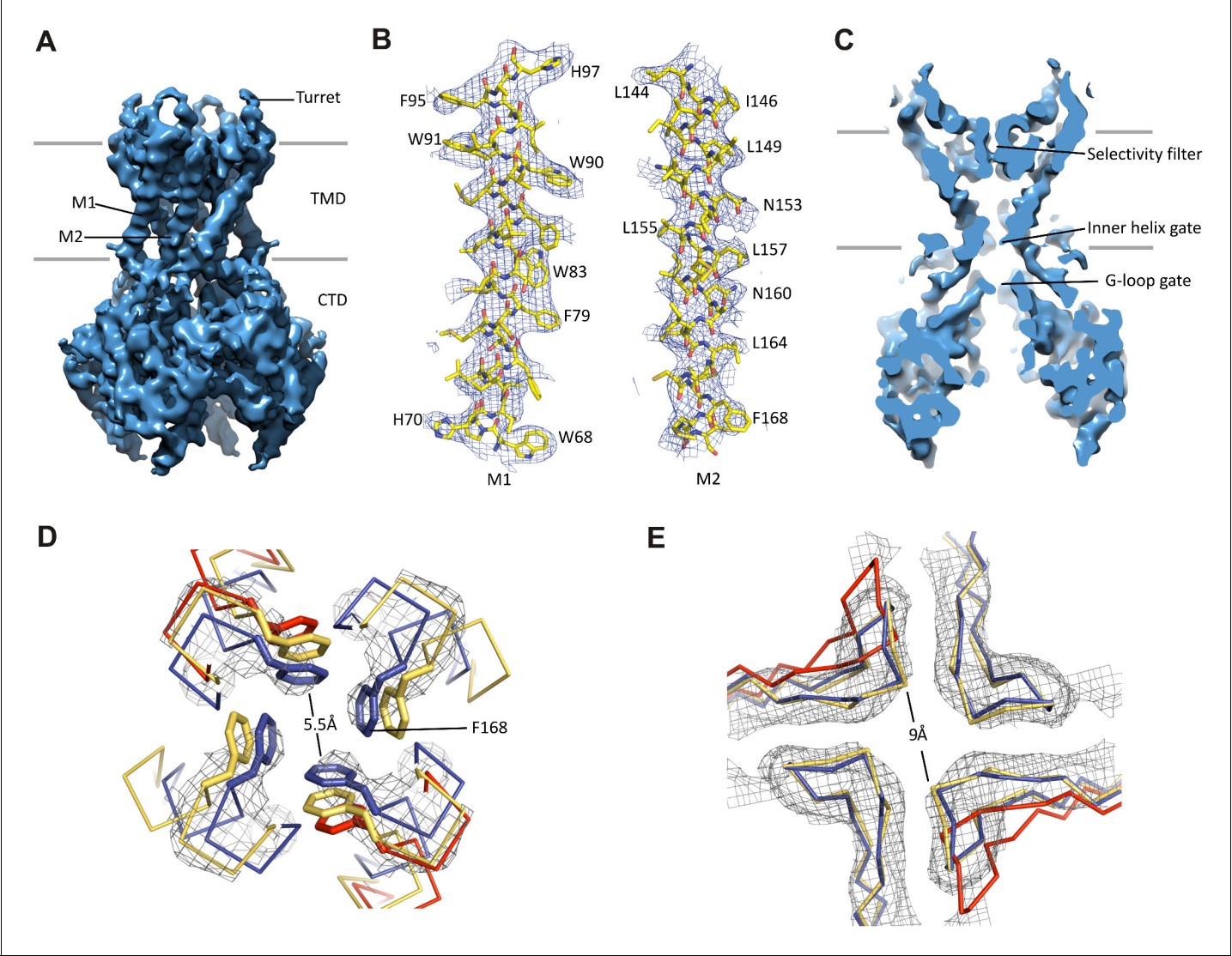

**Figure 3.** Kir6.2 in a closed conformation. (**A**) Cryo-EM density map of Kir6.2 at 5.1 Å resolution. (**B**) Density of M1 and M2. Residues with clear side chain density are labeled. (**C**) A central slice through the density highlighting the ion permeation pathway. (**D**) View of the inner helix gate (F168) looking down the pore from the extracellular side. Kir3.2 apo (yellow, PDB ID: 3SYO) and Kir3.2-R201A+PIP₂ (red, 3SYQ) structures were aligned to the region surrounding the gate. (**E**) Comparison of G-loop conformations of Kir6.2 and Kir3.2 (3SYO and 3SYQ) by alignment of the cytoplasmic domain; same coloring as in (**D**). The distance shown in (**D**) and (**E**) is between the main chains; the constriction should be even narrower due to side chains that should be protruding into the pore, as is seen in homolgous structures. Density depictions contoured to $2.5\sigma$ in (**B**, **D**, **E**).

## Interactions between TMD0-L0 of SUR1 and Kir6.2

As shown in *Figure 2*, TMD0-L0 is sandwiched between the SUR1-ABC core structure and Kir6.2. In the map, densities corresponding to TMD0 and L0 are clearly seen, particularly TMD0, with much of this domain reaching 5 Å resolution. This is in contrast to a recent cryo-EM study of another ABC transporter containing a TMD0, TAP1/2, where TMD0 could not be resolved (*Oldham et al., 2016*), possibly because SUR1-TMD0 in our structure is stabilized by Kir6.2. Overall, TMD0 is a five helix bundle which contains an extracellular N-terminal segment of 25 residues with a brief helical stretch, and mostly short loops connecting helices 2–3, 3–4, and 4–5, but a longer ICL1 of ~14 residues connecting TM1-2 (*Figure 5A*). The N-terminus containing the FLAG-peptide was disordered up until residue C6 of SUR1 where a highly conserved disulfide bond is formed with C26 (*Fukuda et al., 2011*) at the entrance to TM1. This region contacts the Kir6.2 turret and pore loop (*Figure 5—figure*

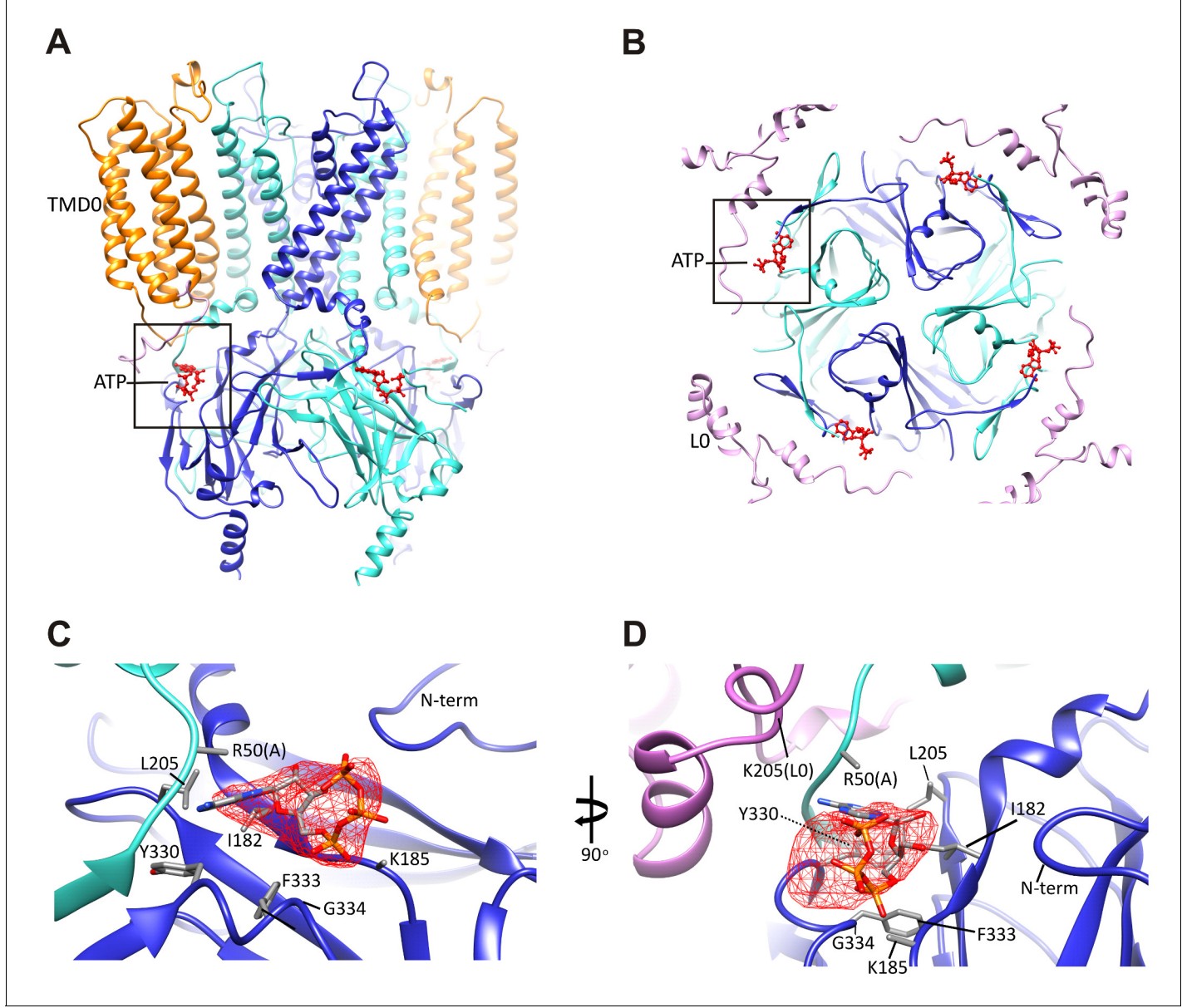

**Figure 4.** The ATP binding pocket. (**A** and **B**) Overview of ATP site from the side and from the top. (**C** and **D**) Difference map calculated from model prior to ATP docking, contoured to 3σ. Residues surrounding the ATP density are labeled. Side chains of residues with supporting density are shown. The N-terminus from Kir6.2 subunit A is colored in cyan and R50 is labeled followed by (**A**). The adjacent subunit is colored in blue, and SUR1-L0 is colored lavender, with the K205 position labeled.

supplement 1A), suggesting a role in assembly and functional coupling with the pore. A number of HI-causing mutations in the N-terminal extracellular loop of TMD0 including C6G, G7R, V21D, N24K, and C26S, which disrupt channel biogenesis efficiency or gating have been reported (*Martin et al., 2016*; *Yan et al., 2007*), further supporting the significance of this region in channel assembly and gating.

In the transmembrane region, TM1 of TMD0 and the M1 helix of Kir6.2 are the primary sites of interaction. These helices make close contact throughout their entire length (*Figure 5A*) and at residue P45 in TM1, a kink is introduced that places the trajectory of the two helices in alignment (*Figure 5—figure supplement 1B*). There are many potential hydrophobic interactions between opposing faces of these helices, which may facilitate association of the complex (*Figure 5—figure*

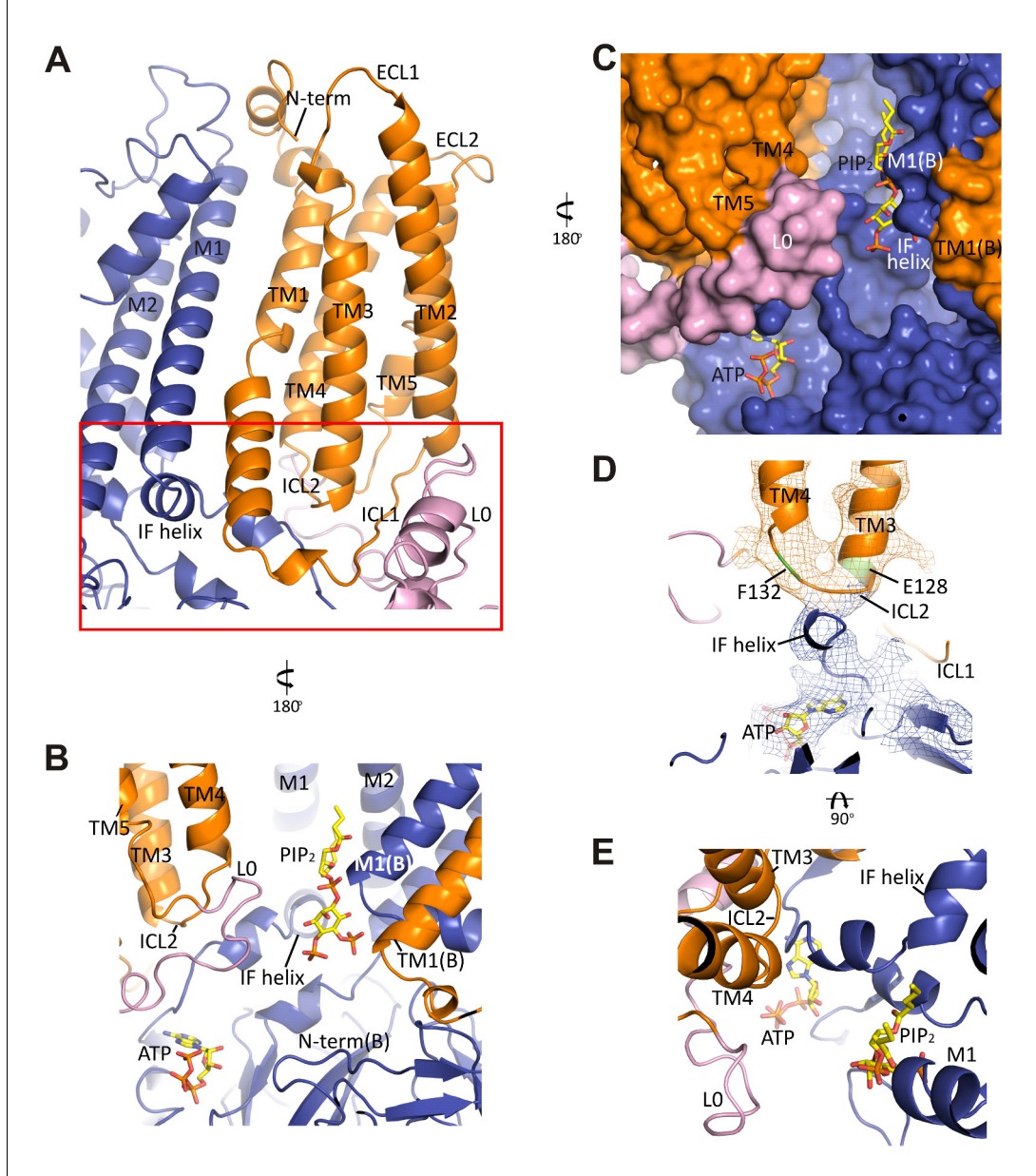

**Figure 5.** The interface between TMD0 and the N-terminal segment of L0 with Kir6.2. (**A**) Overall structure of the interface region, with TMD0 in orange, Kir6.2 in blue, and L0 in lavender. ECL: extracellular loop; ICL: intracellular loop; IF helix: interfacial (slide) helix. (**B** and **C**) Detailed view of the region boxed in red in (**A**) shown in ribbon (**B**) and surface (**C**) representations. ATP is docked as in *Figure 3* and $PIP_2$ was docked hypothetically using $PIP_2$ bound Kir3.2 and Kir2.2 structures for placement. (**D**) A side view of the ICL2 showing close interactions with the Kir6.2 IF helix. E128 and F132, mutation of which alters channel $P_o$ and ATP sensitivity, are highlighted. (**E**) A top-down view of this region with both docked ATP (in the back) and $PIP_2$ in view.
The following figure supplement is available for figure 5:

**Figure supplement 1.** Interactions between TMD0 and Kir6.2.

supplement 1C). Indeed, multiple HI-causing mutations in TM1 of TMD0 (F27S, A30T, L31P, L40R) have been shown to impair channel assembly and surface expression (*Martin et al., 2016*), likely by disrupting interactions between the two helices.

On the cytoplasmic side, there are intimate interactions between the ICLs of TMD0, the start of L0, the Kir6.2 $PIP_2$ binding pocket (cytoplasmic ends of M1 and M2 helices) identified based on

other PIP$_2$-bound Kir structures (*Hansen et al., 2011*; *Whorton and MacKinnon, 2011*), and the Kir6.2 ATP binding pocket. As shown in *Figure 5B and C*, the hypothetically docked PIP$_2$ is surrounded by the cytoplasmic loop connecting TM3 and 4 (ICL2; E128-P133) of TMD0 and the N-terminal stretch of L0 (K192-K199) from one SUR1 subunit, and the cytoplasmic end of TM1 (K57) of TMD0 from the adjacent SUR1 subunit. Previous studies have shown that TMD0 and the N-terminal section of L0 increase the $P_o$ of Kir6.2 to resemble intact channels (*Babenko and Bryan, 2003*; *Chan et al., 2003*). As $P_o$ is determined by PIP$_2$ interactions, our structure suggests these regions may contribute directly to PIP$_2$ binding to account for the increase in PIP$_2$ sensitivity conferred by SUR1 (*Enkvetchakul et al., 2000*). Below PIP$_2$ and near the periphery of Kir6.2 lies ATP, separated from PIP$_2$ by L0 (*Figure 5B,C*) and also ICL2 of TMD0 (*Figure 5B,E*). The ICL2 sits directly atop the Kir6.2 N-terminus, just before the interfacial helix (i.e. the 'slide helix') at Q52 (*Figure 5D*), and simultaneously contacts ICL1 of TMD0 and the most C-terminal portion of TMD0 at TM5. Mutation of E128 (E128K, a HI mutation) and F132 (F132L, a PNDM mutation) in ICL2 as well as Q52 in Kir6.2 (Q52R, a PNDM mutation) is known to disrupt channel gating by ATP and PIP$_2$ (*Pratt et al., 2009*; *Proks et al., 2004*, *2006*) (*Figure 5C,D*). Our finding that this region is close to both the ATP and PIP$_2$ sites illustrates that it is well positioned to contribute to gating regulation by both, explaining the effects of these disease mutations.

## L0 of SUR1 couples the TMD0/Kir6.2 central core to the ABC core of SUR1

L0 (i.e. ICL3) is nestled between TMD0 and the ABC core of SUR1, and comprises ~90 amino acids. We have modeled L0 as a polyalanine chain with two helical segments that are strongly supported by the map, one an amphipathic helix from L224-A240 and the other from L260-D277, which connects to TMD1. In the model, the N- and C-terminal stretches of L0 make a 'V,' with the intervening sequence (L213-L260) forming a hairpin structure at the apex (*Figure 6A,B*). This hairpin structure is simultaneously bridging multiple sites within TMD0 with the ABC core structure (TMs 15 + 16), and may also interact with the Kir6.2 N-terminus (A45-Q52), which would allow L0 to transduce signals from the ABC core to gate the channel. The strategic placement of L0 is consistent with its multiple functional roles reported, including regulation of channel $P_o$, sensitivity to ATP inhibition, and sensitivity to Mg-nucleotide stimulation (*Babenko and Bryan, 2003*; *Chan et al., 2003*; *Masia et al., 2007*).

Another role of L0 that has been reported is interaction with GBC (*Winkler et al., 2012*). GBC is a second-generation sulfonylurea containing a sulfonylurea group and a benzamido moiety that binds K$_{ATP}$ channels with nanomolar affinity (K$_D$ ~1 nM) (*Gribble and Reimann, 2003*). L0 has been proposed to participate in binding to the benzamido group, with mutation Y230A in L0 reducing

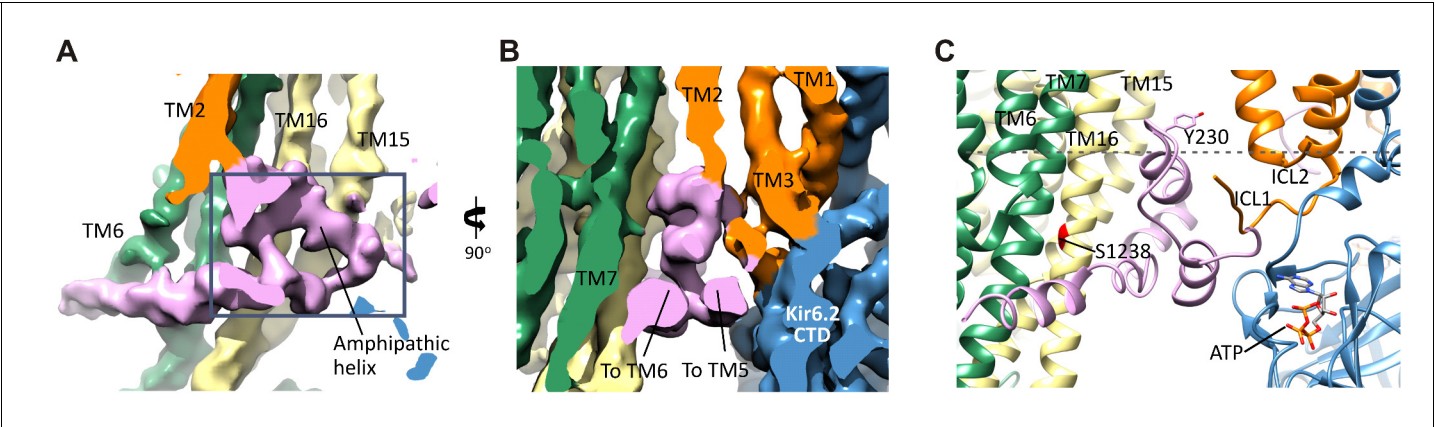

**Figure 6.** The SUR1-L0 connecting TMD0/Kir6.2 with the SUR1-ABC core. (**A**) View of the L0 region from the side along the plane of the membrane; Kir6.2 density has been removed for clarity. The hairpin structure is outlined. (**B**) Slice through the N- and C-terminal segments of L0. (**C**) Model of L0 highlighting relation between Y230 and S1238 (marked red) in TM16, which are separated by ~20 Å (Cα to Cα). Side chain of Y230 is shown based on supporting density. The gray dashed line marks the approximate boundary of the inner leaflet of the lipid bilayer.

GBC binding. We find that the amphipathic helix of L0 containing Y230 sits next to TM16 containing S1238, a residue which when mutated disrupts binding of the sulfonylurea group (*Ashfield et al., 1999*). The two residues are separated by ~20 Å (Cα to Cα), which explains how the two residues distant in the primary sequence can both contribute to binding. Although at the current resolution, we are unable to discern the density for GBC, it is likely to be bound given its high affinity. The model can now be used to guide future studies to clearly define the GBC binding site.

## The SUR1 ABC core in an anomalous inward-facing conformation

The SUR1 core is built from two homologous halves, TMD1-NBD1 and TMD2-NBD2. Each of the 12 combined TM helices from both TMD1 and TMD2 are clearly resolved, as well as the short lateral 'elbow' helices leading into the first helix of each TMD (TM6 and TM12) (*Figure 7A,B*). Characteristic of other ABC exporters, there is a domain swap at the extracellular linker between helices 3 and 4 of each TMD (*Jin et al., 2012*; *Kim et al., 2015*), such that each 'half' of the ABC core is composed of TMs 1–3, and 6 of one TMD, plus TMs 4 and 5 of the other (*Figure 7A*).

Overall, the SUR1-ABC core is in an inward-facing conformation, with the NBDs clearly separated (*Figure 7C*). This is consistent with other ABC exporter structures solved without Mg-nucleotides. However, in contrast to other ABC exporters of known structure whereby transporter halves are related by either a true or a pseudo two-fold symmetry axis, depending on whether the two halves are identical or not (*Wilkens, 2015*), we find a clear rotation and a translation of TMD1-NBD1 relative to TMD2-NDB2, such that TMD1-NBD1 is ~15° off the symmetry axis and is translated by ~10 Å horizontally (relative to the membrane) (*Figure 7C*). In this configuration, the SUR1 NBDs likely could not dimerize without a twisting motion to align the dimerization interface.

Dimerization of NBDs in SUR1 has been proposed to follow MgATP hydrolysis or MgADP binding to stimulate channel activity (*Nichols, 2006*), and GBC inhibits channel activity by preventing Mg-nucleotide stimulation (*de Wet and Proks, 2015*; *Gribble and Reimann, 2003*). As discussed above, given its high-affinity GBC is likely to be bound in our structure. Thus, an interesting hypothesis is that the twisted conformation is caused by GBC binding, which would suggest that GBC prevents MgADP from stimulating the channel by causing a misalignment of the NBDs dimerization interface. Alternatively, the conformation may be unique to SUR1 and that Mg-nucleotide binding/hydrolysis is required to restore symmetry for dimerization. In this case, GBC may block stimulation by clamping down L0 and preventing it from communicating with Kir6.2. A structure in the absence of GBC will be needed to test these hypotheses.

## Discussion

The structure reported here provides the first glimpse of the detailed domain organization of $K_{ATP}$ channels and the intricate structural interactions between SUR1 and Kir6.2. These data offer mechanistic insight into how SUR1 and Kir6.2 function as a complex to regulate insulin secretion (*Figure 8A*). We propose that like other ABC transporters (*Wilkens, 2015*), the ABC core of SUR1 switches between an inward-facing and outward-facing conformations as MgATP undergoes hydrolysis or as MgADP binds at NBD2 and induces NBD dimerization. The conformational switch at the ABC core causes movement of the L0 and TMD0, which alters channel interactions with ATP and $PIP_2$ by remodeling the interface formed by the cytoplasmic domain of Kir6.2, the bottom of the Kir6.2 transmembrane helices, the intracellular loops of TMD0 and the N-terminal segment of L0. In this way, the SUR1 'transport' cycle is coupled to Kir6.2 opening or closing rather than transport of substrates through SUR1 itself.

Our structure highlights the critical role of SUR1-TMD0 in the association of the two subunits. In addition to contacts made by TM1 of Kir6.2 and the first TM helix of TMD0 which are consistent with previous structure-function studies (*Schwappach et al., 2000*), there are also new interactions revealed by the structure in the extracellular domain of TMD0 and the turret/pore loop of Kir6.2 as well as the cytoplasmic domains of TMD0 and Kir6.2. Indeed, TMD0 appears to harbor more mutations that disrupt channel biogenesis and trafficking than other regions of SUR1 (*Martin et al., 2013*, *2016*). It is worth noting that many mutations in TMD0 which impair channel biogenesis and trafficking can be rescued by pharmacological chaperones, specifically sulfonylureas such as GBC (*Chen et al., 2013a*; *Martin et al., 2016*; *Yan et al., 2004*, *2007*). As our structure is obtained in the presence of GBC, an important question to address in the future is whether GBC alters structural

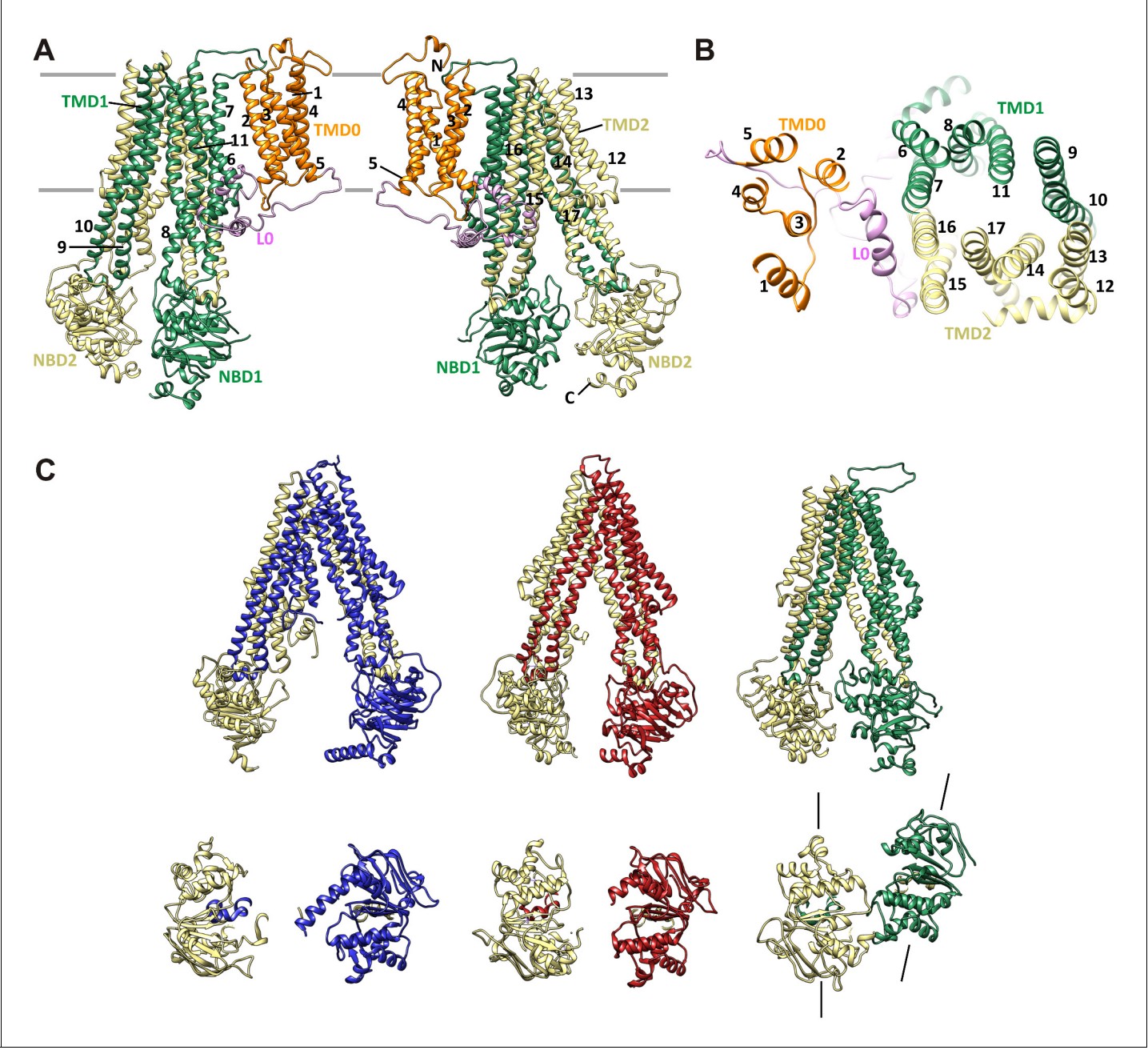

**Figure 7.** SUR1 with a twisted ABC core conformation in saturating concentrations of GBC. (**A**) Model of SUR1 with the various domains colored as in *Figure 1*, with each TM helix labeled. On the left, TMD1/NBD1 (green) is toward the front and TMD2/NBD2 (tan) is toward the back. (**B**) Cross-section of the SUR1 model, showing relative orientation of each of the 17 TM helices and a helix in L0. (**C**) Comparison of inward-facing ABC transporter structures: From left, *C elegans* Pgp (PDB code 4 F4C); mouse Pgp (4 M1M); hamSUR1. For each model, TMD2/NBD2 is colored tan. Lines on the side of the SUR1 NBDs denote the relative orientation of the NBD dimerization interface, demonstrating the observed twisting relative to other inward-facing structures.

interactions between TMD0 and Kir6.2 to correct biogenesis/trafficking defects caused by TMD0 mutations.

The interface between TMD0-L0 and Kir6.2 in the cytoplasmic domain near the proposed $PIP_2$ binding site and where ATP density is observed suggests TMD0-L0 may directly enforce $PIP_2$ or ATP binding to enhance Kir6.2 sensitivity to both, and also explains the effects of many disease mutations

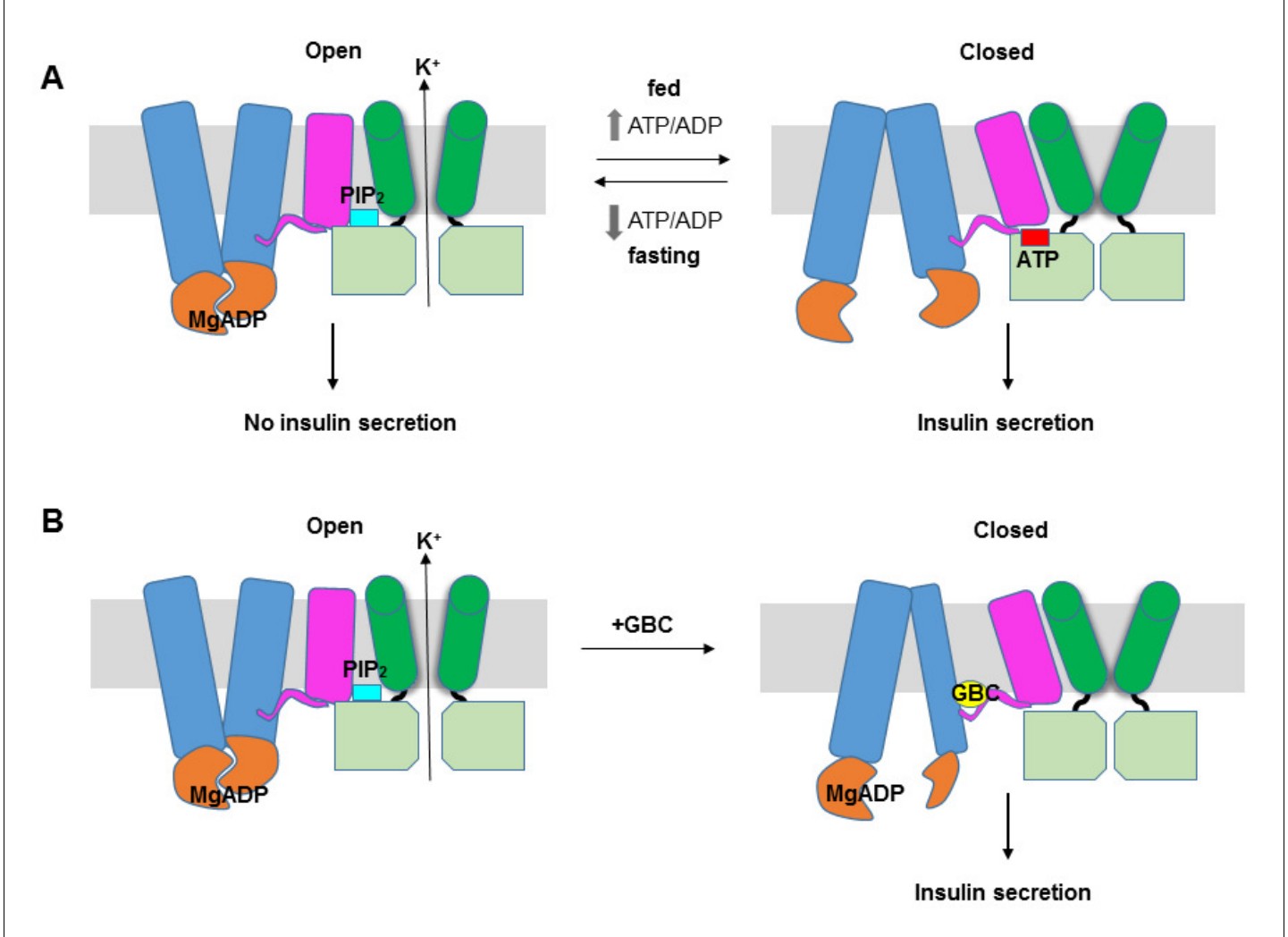

**Figure 8.** $K_{ATP}$ channel gating model. (**A**) Cartoon illustrating how changes in the ATP/ADP ratio upon feeding and fasting alter the equilibrium between the inward-facing and outward-facing states of the SUR1-ABC core and interactions of the channel with ATP and PIP$_2$ to control channel activity. (**B**) Model of the hypothesized mechanism whereby GBC causes misalignment of the NBDs to prevent Mg-nucleotides activation of $K_{ATP}$ channels. In both **A** and **B**, Kir6.2 transmembrane helices: green; Kir6.2 cytoplasmic domain: lime green; SUR1-TMD0/L0: magenta; SUR1-TMD1/2: blue; SUR1-NBDs: orange; GBC: yellow; ATP: red; PIP$_2$: cerulean. Note the different states shown are not meant to reflect the actual conformational transitions.

in this region. Although in our structure the Kir6.2 is bound to ATP with the pore in a closed conformation, a gating scheme whereby in the presence of PIP$_2$ remodeling of the interfaces near the ATP and PIP$_2$ sites leads to channel opening may be envisioned. Future studies comparing structures in the absence of ATP and with or without PIP$_2$ are needed to understand in detail the structural changes involved in gating.

The L0 region before the elbow helix leading to TMD1 in SUR1 was modeled de novo, with an amphipathic helix from L224-A240 and a helix from L260-D277 that are strongly supported by the density map. Part of L0 (from a.a. 214 on) is conserved in CFTR and the multidrug resistance-associated proteins MRPs (*Zhang and Chen, 2016*). Interestingly, in the recently reported CFTR structure, this loop which the authors named the 'lasso motif' also contains an amphipathic helix followed by another helix before the elbow helix (*Zhang and Chen, 2016*). Our structural model of L0 is in line with the CFTR model of the corresponding loop. In CFTR or MRP-1, this loop has been shown to be involved in trafficking regulation by syntaxin 1A (*Naren et al., 1998*; *Peters et al., 2001*) or

association with the plasma membrane (*Bakos et al., 2000*), respectively. It would be interesting to determine whether L0 of SUR1 has similar roles.

A striking feature observed in our structure is the unexpected twisted inward-facing conformation of the SUR1-ABC core that is distinct from other ABC transporter apo-state structures (*Wilkens, 2015*). This observation suggests a possible mechanism in which GBC inhibits channel activity by preventing dimerization of NBDs in the presence of Mg-nucleotides (*Figure 8B*). As GBC is known to inhibit the activity of other ABC transporter proteins including CFTR (*Schultz et al., 1996*) and the multidrug resistance protein MDR (*Golstein et al., 1999*), the mechanism we propose could have broader implications. Intriguingly, close examination of the recently published zebrafish CFTR structure where the inhibitory R-domain is present (*Zhang and Chen, 2016*) and the TAP transporter structure with an inhibitory viral peptide bound (*Oldham et al., 2016*) also indicates misalignment of the two NBDs albeit to lesser degrees, further suggesting that NBDs misalignment may be a common theme in ABC transporters bound to inhibitory ligands.

In summary, the novel insight gained from our structure lays the foundation for future structural and functional studies. In particular, structures bound with various stimulatory and inhibitory ligands will further advance understanding of the detailed mechanisms of channel gating. Some regions known to be important for channel assembly and gating such as the distal N- and C-termini of Kir6.2 as well as several linker loops in SUR1 are not well resolved in the current map (see Materials and methods for details). An equally important future goal is to stabilize these regions and obtain higher resolution structures to fully visualize the channel.

## Materials and methods

### Construction of recombinant adenoviruses

Construction of the hamster SUR1 (94.5% protein sequence identity with human SUR1) with an N-terminal FLAG-tag (f-SUR1) and rat Kir6.2 (96.15% protein sequence identity with human Kir6.2) recombinant adenoviruses was as described previously (*Lin et al., 2005*; *Pratt et al., 2009*). A FLAG tag (DYKDDDDK) was engineered at the N-terminus of SUR1 for affinity purification of the channel complex. In brief, the gene encoding the rat Kir6.2 was cloned into pShuttle, and recombined with the pAdEasy vector in the BJ5183 strain of *Escherichia Coli*. Positive recombinants were selected, and pAdEasy plasmids containing the correct insert were used to transfect HEK293 cells (RRID: CVCL_0045) for virus production. The SUR1 recombinant adenovirus was constructed using a modified pShuttle plasmid (AdEasy kit, Stratagene, San Diego, CA) containing a tetracycline-inducible promoter. Recombinant viruses were amplified in HEK293 cells and purified according to the manufacturer's instructions.

### K$_{ATP}$ channel expression and purification

INS-1 cells clone 832/13 (RRID:CVCL_7226) (from Dr. Christopher Newgard) (*Hohmeier et al., 2000*) were plated in 15 cm plates and cultured for 24 hr in RPMI 1640 with 11.1 mM D-glucose (Invitrogen, Carsbad, CA) supplemented with 10% fetal bovine serum, 100 units/ml penicillin, 100 μg/ml streptomycin, 10 mM HEPES, 2 mM glutamine, 1 mM sodium pyruvate, and 50 μM $\beta$-mercaptoethanol. For channel expression, cells were co-infected with three recombinant adenoviruses, one encoding Kir6.2, one f-SUR1, and one encoding tetracycline-inhibited transactivator (tTA) for the tTA-regulated f-SUR1 expression (*Pratt et al., 2009*). Cells at ~70% confluent density were washed once with phosphate-buffered saline (PBS) and then incubated for 3 hr at 37°C in OPTI-MEM without serum and a mixture of viruses with the multiplicity of infection (M.O.I.) of each virus determined empirically to optimize the maturation efficiency of the channel complex as judged by the abundance of the SUR1 and Kir6.2 bands as well as the ratio of the mature complex glycosylated versus the immature core-glycosylated SUR1 bands. Medium was then replaced with fresh growth medium plus 1 mM sodium butyrate and 1 μM glibenclamide (GBC) to enhance expression and maturation (*Yan et al., 2004*), and the cells were further incubated at 37°C for 36–48 hr. Cells were harvested in PBS, pelleted, flash frozen in liquid nitrogen, and stored at −80°C until purification.

For channel purification, cells were resuspended in hypotonic buffer (15 mM KCl, 10 mM HEPES, 1.5 mM MgCl$_2$) and allowed to swell for 20 min on ice. Cells were then lysed with a tight-fitting Dounce homogenizer, then centrifuged at 20,000 x*g* for 60 min. Membranes were resuspended in

buffer A (150 mM NaCl, 25 mM HEPES, 50 mM KCl, 1 mM ATP, 1 µM GBC, 4% Trehalose) with protease inhibitors (cocktail tablets from Roche) and then solubilized with 0.5% Digitonin for 90 min. Solubilized membranes were separated from insoluble materials by centrifugation (100,000 x$g$ for 30 min at 4°C) and then incubated with anti-FLAG M2 affinity agarose gel for 4–5 hr. The protein-bound agarose gel was washed with five column volumes of buffer B (150 mM NaCl, 25 mM HEPES, 50 mM KCl, 1 mM ATP, 1 µM GBC, 0.05% Digitonin) and bound proteins eluted in the same buffer with FLAG peptide. Eluted proteins were concentrated using a centricon filter (100 kD cutoff) to a final concentration of ~0.7–1 mg/ml. Purified proteins were further fractionated by size exclusion chromatography using a Suprose six column and fractions analyzed by blue native gel electrophoresis and SDS-PAGE (*Figure 1A,B*).

## Sample preparation and data acquisition for cryo-EM analysis

Digitonin solubilized $K_{ATP}$ complexes (in the presence of 1 mM ATP and 1 µM GBC) were first examined by negative-staining EM (1% w/v uranyl acetate, on continuous thin-carbon coated grids) to confirm the integrity of the full complex (*Figure 1C*). For cryo-EM imaging, due to low particle distribution with holey-carbon grids, we experimented with two types of grids: UltrAufoil gold grids and C-flat grids coated in-house with 5 nm of gold on each side, and used both in the final data collection. The grids were first glow-discharged by EasyGlow at 20 mA for 45 s, then 3 µl of purified $K_{ATP}$ complex was loaded onto the grid, blotted (2–4 s blotting time, force −4, and 100% humidity) and cryo-plunged into liquid ethane cooled by liquid nitrogen using a Vitrobot Mark III (FEI, Hillsboro, OR).

Single-particle cryo-EM data weres collected on a Titan Krios 300 kV cryo-electron microscope (FEI) in the Multi-Scale Microscopy Core at Oregon Health and Science University, assisted by the automated acquisition program SerialEM. Images were recorded on the Gatan K2 Summit direct electron detector in the counting mode at the nominal magnification 81,000 x (calibrated image pixel-size 1.720 Å), with varying defocus ranging between 1.2 and 3.5 µm across the dataset (*Figure 1D*). To contain the beam radiation damage and reduce electron coincidence loss in the K2 counting-mode recording, the dose rate was kept around 2.0 e⁻/Å²/s, frame rate at 2 frames/s and 40 frames in each movie, which gave the total dose of approximately 40 e⁻/Å². In total, 4339 movies were recorded, from which ~35,000 particles were used in final reconstructions (*Figure 1—figure supplements 1* and *2*).

## Image processing

The raw frame stacks were gain-normalized and then aligned and dose-compensated using Unblur (*Grant and Grigorieff, 2015*) (*Table 1*). CTF was estimated from the aligned frame sums using CTFFIND4 (*Rohou and Grigorieff, 2015*). To reduce the possibility of bias and capture every possible particle view, an initial set of 350,000 potential particles (referred to as 'peaks' in *Figure 1—figure supplement 1*) were picked using DoGPicker (*Voss et al., 2009*) with a broad threshold range for subsequent 2D classification using RELION (*Scheres, 2012*). 2D classification was able to remove the large number of false positives and aggregates, and resulted in ~35,000 particles with 2D classes in which secondary structure was already apparent (*Figure 1E*). These class averages revealed that the side views also adopted a preferred orientation. Upon imposing C4 symmetry, the angular sampling space was filled in along three orthogonal axes (*Figure 1—figure supplement 2A*), which greatly improved the quality of the 3D reconstruction. The final rounds of refinement with C4 symmetry revealed two 3D classes (*Figure 1—figure supplement 1*). The dominant class (EMDB ID: EMD-8470), derived from 20,707 particles had an overall resolution of ~6.7 Å, and application of a mask improved the resolution of the overall structure to 5.8 Å and the central Kir6.2 domain to 5.1 Å (*Figure 1—figure supplement 2B*). The second class, derived from 14,115 particles, had an overall unmasked resolution of ~7.6 Å, and masking improved the resolution of the overall structure to 7.2 Å and for the central Kir6.2 domain to 6.9 Å (*Figure 1—figure supplement 2C*). All resolutions were reported using the 0.143 criterion with gold-standard FSC and phase-randomization correction for the use of masks (*Chen et al., 2013b*). Resolution was further confirmed using local-resolution as measured using ResMap (*Kucukelbir et al., 2014*), and by observing criterion such as helical pitch starting to become visible, and density bumps for some of the larger side chains (see examples shown in *Figure 3B*). Maps were B-factor corrected during post-processing using the K2 MTF, and

the fitting procedure described by Rosenthal and Henderson (*Rosenthal and Henderson, 2003*). The two 3D classes differ in the cytoplasmic domain of Kir6.2 where a rotation of ~14° relative to each other was observed (*Figure 1—figure supplement 2F*).

## Model building

Local resolution measurements using ResMap and masked FSCs showed that some parts of the complex including Kir6.2 and TMDs of SUR1 had significantly better resolution, in the 5 Å range, than the overall resolution of 5.8 Å, while other parts such as the NBDs of SUR1 had worse resolution, estimated to be in the 8 Å range. Moreover, some parts of the channel complex, such as the TMD0 and L0 of SUR1 do not have existing homology models. Therefore, different strategies were used to model the channel complex, as detailed below.

For Kir6.2, a homology model was built from Kir3.2 (PDB ID: 3SYA) using MODELLER (*Webb and Sali, 2016*) and served as the initial model. The model was docked into the density in UCSF Chimera (*Pettersen et al., 2004*); the fit was improved by rigid body refinement of domains in RSRef (*Chapman et al., 2013*), followed by iterative rounds of real-space refinement in COOT (*Emsley et al., 2010*) and stereochemically restrained torsion angle refinement in CNS (*Brünger et al., 1998*), substituting in the RSRef real-space target function (*Chapman et al., 2013*), adding (φ,ψ) backbone torsion angle restraints, and imposing non-crystallographic symmetry (NCS) constraints. The final model contained residues 32–356 (*Figure 2—figure supplement 1*). The distal N- and C-termini of Kir6.2, although interesting regions implicated in channel assembly and gating (*Devaraneni et al., 2015*; *Enkvetchakul et al., 2000*; *Zerangue et al., 1999*), lacked strong density. Therefore, they were not included in the model. For the SUR1 core structure, the sequence was divided into three segments: TMD1, NBD1, and TMD2-NBD2. A TMD1 homology model was built using PCAT-1 (PDB ID: 4RY2) (*Figure 2—figure supplement 2*), NDB1 was modeled from the NDB1 of mouse P-glycoprotein (PDB ID: 4 M1M) (*Figure 2—figure supplement 3*), and TMD2 and NBD2 were modeled together from chain B of TM287/288 (PDB ID: 4Q4HB) (*Figure 2—figure supplement 4*); all homology models were built with MODELLER. These models were docked into the density in Chimera.

SUR1 had some disordered regions (744–770, 928–1000, 1319–1343), particularly in the linkers between TMDs and NBDs, and in NBD1, that were not seen in our map. These regions were removed from the homology models before proceeding with refinement. The TM helices were then manually adjusted in COOT, as a substantial adjustment was needed to move them into density. The domains were then refined in the same steps as outlined for Kir6.2, except that before the final manual adjustments in COOT and final density gradient optimization, a batch of torsion angle simulated annealing optimization was inserted, again using RSRef/CNS and the same torsion angle restraints and NCS constraints. The final model for the ABC core structure contained residues 284–616 (TMD1), 675–739 and 762–930 (NBD1), 981–1044 and 1060–1321 (TMD2), and 1325–1577 (NBD2).

TMD0 and L0 domains of SUR1 (a.a. 1–283) are some of the most interesting and novel regions of the $K_{ATP}$ complex for which there is no existing homology model. These domains were therefore modeled de novo. Even though embedded in a micelle, all the transmembrane helices in TMD0 are clearly visible in the density map. The visibility of helical pitch and some side chains allowed confident modeling and refinement of the TM helices. With the predominantly alpha-helical nature of this domain, continuous loop density between most of the TM helices, and the presence of residues with bulky side chains, we were able to build the ~200 residues of TMD0 with a good degree of confidence. Of less certainty was the L0 region of SUR1 that sits between TMD0 and TMD1. While there was an easily identifiable region of the map corresponding to L0, the scarcity of secondary structures in this region made it difficult to build with the same degree of confidence. This was further complicated by the high likelihood that some of the observed density may be attributable to the ligand GBC, a high affinity antagonist which has been shown to interact with this region (*Bryan et al., 2004*). Nonetheless, we made a best effort to model the residues in L0 primarily to verify that (1) a plausible model could be built into this density, and (2) that the observed density was sufficient to account for all the amino acids in this loop. The L0 model we built fulfilled both criteria, and as such, allowed for a better interpretation and understanding of the electron density map. We did not, however, attempt to draw any definitive conclusions about specific residues or GBC density from our tentative modeling of L0.

Note we used two different software suites, RSRef and PHENIX (*Adams et al., 2010*), to confirm the consistency of our individual models of Kir6.2 and the SUR1 ABC core structure upon refinement into our electron density. The full final models were refined with all the constraints available in PHENIX real-space refinement: torsion angles, bond lengths, Ramachandran, and secondary structure. This was done initially with side-chains in place to ensure that the refinement did not place residues in implausible configurations (*Figure 3B* shows examples of residues that were particularly well-resolved and served as anchor points for building and refining the model). Evaluation of these refined models confirmed that the model could be refined to fit the density quite well while maintaining good stereochemical statistics (*Table 1*). However, as many of the side chains did not have much, if any, supporting density, a final pass was made throughout the entire model to remove these side-chains prior to PDB deposition (PDB ID: 5TWV). The resulting model was very similar to the full-atom refinement, but had better statistics (*Table 1*) primarily due to the reduced possibility of clashes.

## Acknowledgements

We thank Nicholas Caputo for assistance with microscope operation and data collection, Dr. Omar Davulcu and Dr. Michael Chapman for assistance on structure refinement using RSRef. We are grateful to Dr. Eric Gouaux and Dr Michael Chapman for helpful discussion and comments on the manuscript. We also thank the staff at the Multiscale Microscopy Core (MMC) of Oregon Health and Science University (OHSU), the OHSU-FEI living lab and Intel for technical support. This work was supported by the National Institutes of Health grants R01DK066485] (to S-L S) and F31DK105800 (to GMM).

## Additional information

### Funding

| Funder | Grant reference number | Author |
| --- | --- | --- |
| National Institute of Diabetes and Digestive and Kidney Diseases | R01DK066485 | Show-Ling Shyng |
| National Institute of Diabetes and Digestive and Kidney Diseases | F31DK105800 | Gregory M Martin |

The funders had no role in study design, data collection and interpretation, or the decision to submit the work for publication.

### Author contributions

GMM, Data curation, Formal analysis, Funding acquisition, Validation, Investigation, Visualization, Methodology, Writing—original draft, Writing—review and editing; CY, Data curation, Software, Formal analysis, Validation, Visualization, Writing—review and editing; EAR, Investigation, Methodology, Project administration; JFF, Formal analysis, Investigation, Methodology; QX, Software, Validation, Writing—review and editing; MRW, Formal analysis, Methodology, Writing—review and editing; JZC, Data curation, Supervision, Funding acquisition; S-LS, Conceptualization, Resources, Data curation, Formal analysis, Supervision, Funding acquisition, Investigation, Methodology, Writing—original draft, Project administration, Writing—review and editing

### Author ORCIDs

Show-Ling Shyng, http://orcid.org/0000-0002-8230-8820

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
