## [Decision Letter]

Thank you for submitting your article "Cryo-EM structure of K_ATP_ channels illuminates mechanisms of assembly and gating" for consideration by *eLife*. Your article has been favorably evaluated by John Kuriyan (Senior Editor) and three reviewers, one of whom, Werner Kühlbrandt (Reviewer #1), is a member of our Board of Reviewing Editors. The following individuals involved in review of your submission have agreed to reveal their identity: John L Rubinstein (Reviewer #2); Yifan Cheng (Reviewer #3).

The reviewers have discussed the reviews with one another. They agree that this is a landmark paper and recommend rapid publication. The Reviewing Editor has drafted this decision letter to help you prepare a revised submission.

Summary:

Martin and colleagues present the structure by cryo-EM of the ATP-sensitive potassium channel at ~6 Å resolution in the presence of ATP and glibenclamide. This is a landmark structure of fascinating, complicated, and an extremely (and increasingly!) biomedically-important macromolecular assembly. The structural analysis is well done and the depth of expertise for the biological interpretation of the structure is readily apparent. Insight is gained into numerous points, including where ATP and Glibenclamide bind the K_ATP_ channel and how they affect channel function. As a particularly interesting aspect, the structure illustrates how information about ATP concentrations in the cell, sensed by the ABC protein SUR1 subunits, is communicated to the Kir6.2 channel by TMD0 and the L0 linker.

Nevertheless, the reviewers felt that some revisions of the text are advisable, to make it more accessible to a broad readership.

Recommended revisions:

The manuscript is rich in detailed description of various interactions, but less so in mechanistic insights. An obvious question is how does an ABC transporter facilitate the gating of an ion channel? We realize that this question cannot be fully addressed on the basis of a single structure, but a more detailed discussion would be appropriate.

The ATPase activity of SUR1 is not mentioned. Is the activity known? If so, please cite the relevant literature, unless you have measured it yourself – in which case please include it.

Do the SUR subunits by themselves have any transport activity? This interesting question should be discussed and suitable references provided.

Is the density assigned to bound ATP present in both conformations or only in one?

Is it possible that it might be ADP? A reference that supports the assumption that it is indeed bound ATP would be helpful.

Considering that PIP2 is required for Kir channel functionality, is it possible that PIP2 has dropped off during isolation (which would be one explanation why it is not visible in the map), and that this is the reason why the channel is locked in a closed state, rather than because of ATP binding and drug inhibition?

---

## [Author Response]

*Recommended revisions:*

*The manuscript is rich in detailed description of various interactions, but less so in mechanistic insights. An obvious question is how does an ABC transporter facilitate the gating of an ion channel? We realize that this question cannot be fully addressed on the basis of a single structure, but a more detailed discussion would be appropriate.*

Following the reviewers’ suggestion, we have now added a cartoon figure and a paragraph in Discussion to propose a possible mechanism by which SUR1 regulates Kir6.2 gating based on our structure presented in the paper and data in the literature.

*The ATPase activity of SUR1 is not mentioned. Is the activity known? If so, please cite the relevant literature, unless you have measured it yourself – in which case please include it.*

The ATPase activity of SUR2A and recombinant NBD2 of SUR1 have been reported in the literature. We have now cited these studies and added a statement about the current model of how ATPase activity of the NBD2 of SUR1 is coupled to Kir6.2 gating in Introduction.

*Do the SUR subunits by themselves have any transport activity? This interesting question should be discussed and suitable references provided.*

To our knowledge, no transport activity has been reported for SUR1 or SUR2. In fact, the SUR proteins have often been referred to as silent ABC transporters whose sole known functions are to regulate the activity of Kir6.2 (or Kir6.1). We have now added a sentence to address this question with suitable references in Introduction.

*Is the density assigned to bound ATP present in both conformations or only in one?*

*Is it possible that it might be ADP? A reference that supports the assumption that it is indeed bound ATP would be helpful.*

We did not observe significant difference in the “ATP” density between the two states, which suggests the rotational difference in the CTD of Kir6.2 between the two classes may stem from inherent flexibility of the CTD. However, it remains a possibility that the motion is part of a conformational transition between ATP-bound and ATP-unbound states. In our preparation, ATP was present at saturating concentrations, which may prevent complete transition from the ATP-bound state to the ATP-free state. Moreover, the presence of GBC may also prevent complete transition to a second state via interaction between the CTD of Kir6.2 and the TMD0/L0 of SUR1. We decided to note the presence of this second state both because it is an interesting observation, and also because separating out these particles was necessary to reach the higher resolution in the better map. There is likely continuous heterogeneity between the high resolution map and the second lower-resolution map, and there may be more 3-D classes to be distinguished, but this was the best result we could achieve with our current dataset size. As stated in the manuscript, a structure in the absence of ATP in the future will be needed to address these questions. We have now revised the Results section to make it clear that ATP density was present in both classes.

As to the question of whether the density we observed is ADP, in our preparation only 1mM ATP was included without Mg^2+^. This makes it unlikely that the density we observed was ADP. Biochemical studies using azido-ATP have provided direct evidence that ATP binds to Kir6.2. Moreover, Kir6.2 channels formed in the absence of SUR1 (using a Kir6.2 variant lacking the RKR ER retention signal in the C-terminus) are inhibited by ATP as well as non-hydrolyzable ATP analogues, consistent with direct interaction of Kir6.2 with ATP. These studies, cited in the paragraph discussing the ATP binding site in Kir6.2, support our assumption and interpretation of the bound ATP density. We have now also included these background studies in Introduction to improve clarity.

*Considering that PIP2 is required for Kir channel functionality, is it possible that PIP2 has dropped off during isolation (which would be one explanation why it is not visible in the map), and that this is the reason why the channel is locked in a closed state, rather than because of ATP binding and drug inhibition?*

It is possible that PIP_2_ is present in our digitonin-solubilized sample preparations. However, at the resolution we have it is difficult to confidently identify PIP_2_ density, especially along the micelle surface. This also applies to glibenclamide. In interpreting the structural data, we try to be rigorous and believe that higher resolution is needed to resolve PIP_2_. With regard to the closed Kir6.2 conformation observed, ATP and glibenclamide are two potent channel inhibitors even with physiological PIP_2_ present (IC_50_~10µM and 1nM respectively measured in inside-out patch-clamp recordings). We have now revised the text concerning PIP_2_ to make this point clear.